# Study on Urban Resilience from the Perspective of the Complex Adaptive System Theory: A Case Study of the Lanzhou-Xining Urban Agglomeration

**DOI:** 10.3390/ijerph192013667

**Published:** 2022-10-21

**Authors:** Liangjie Yang, Hainan Yang, Xueyan Zhao, Yongchun Yang

**Affiliations:** 1College of Geography and Environment Science, Northwest Normal University, Lanzhou 730070, China; 2Key Laboratory of Resource Environment and Sustainable Development of Oasis, Lanzhou 730070, China; 3College of Earth and Environmental Sciences, Lanzhou University, Lanzhou 730030, China

**Keywords:** complex adaptive system theory, evolutionary resilience, adaptive cycle model, urban system, Lanzhou-Xining urban agglomeration

## Abstract

In the context of global environmental change and continuous urbanization, enhancing urban resilience is an important way to improve urban emergency management capacity and achieve sustainable development of urban systems. It is of great significance to clarify the mechanisms and effects of urban resilience and carry out resilience measurement to improve the level of urban system resilience and alleviate the pressure of environmental disturbances on the stable operation of urban systems. As an important part of the “Belt and Road” Initiative and one of the few leading economic regions in western China, promoting the high-quality development of the Lanzhou-Xining urban agglomeration is of profound significance for strengthening ethnic unity and stabilizing the northwest and southwest regions. Based on the complex adaptive system (CAS) theory and the adaptive cycle model, this study understands urban resilience as the comprehensive result of urban system stability, self-organization, learning adaptability and transformability, constructs a multi-level open index evaluation system, and analyzes the spatio-temporal evolution characteristics of urban resilience of the Lanzhou-Xining urban agglomeration from the proposed design to the formal planning in 2010–2017. The findings are as follows: (1) Research on the urban resilience of the Lanzhou-Xining urban agglomeration verifies the applicability of the evolutionary urban resilience analysis framework and makes preliminary findings on urban resilience based on CAS theory, which provide a certain theoretical reference for the research on the spatio-temporal evolution of urban resilience. (2) From 2010 to 2017, significant differences are observed between various urban attributes. Resilience exhibits an overall upward trend, and spatial evolution changes from a double core (Lanzhou and Xining) to three cores (Lanzhou, Xining and Haidong) and polycentric modes. (3) Based on urban resilience characteristics and an urban system adaptability cycle model, this paper divides the Lanzhou-Xining urban agglomeration cities into four types (exploitation–reorganization, conservation–release, conservation–exploitation and exploitation), and proposes corresponding adaptive management countermeasures. These could be adopted as a reference to promote the high-quality development of the Lanzhou-Xining urban agglomeration.

## 1. Introduction

A city is composed of natural ecosystems, artificial building systems and human social systems. It is constantly affected by external and internal urban system pressures, such as extreme weather conditions, geological disasters, large-scale economic crises and public health emergencies. The global population is approximately 7.6 billion, and it is estimated that by 2050, the population living in urban areas will increase from 55% to 68% [1]. The urbanization trend is rapid, but not all cities have a benign development trend. In the face of various pressures, certain cities, such as Tokyo, Japan, which was rebuilt after World War II, and Tangshan, China, which recovered and developed after an earthquake, may be influenced by these shocks and pressures to stimulate their potential for further development along a better path. However, other cities may become stagnated or even disappear due to excessive external shocks or unbalanced internal structural adjustments, such as Pompeii, Italy, which disappeared after a volcanic eruption, and Loulan, China, which disappeared after excessive human disturbance to the ecological environment. Therefore, cities must become more resilient and smarter in response to a series of challenges in the process of urban development by flexibly and actively enhancing their resistance and adaptability [2].

In the process of urban system evolution, each subsystem has a different scale, direction and speed, which leads to system diversity and complexity [3]. On the one hand, with increasing urban system complexity, the urban structure becomes increasingly effective and stable, which promotes sustainable development [4]. On the other hand, the complexity of cities also leads to increasingly uncertain security problems [5]. Therefore, the study of system complexity has become an important topic in current urban research. In the study of urban resilience, many scholars have also noted this problem [6,7].

Resilience studies outside China were conducted early on. Scholars and organizations such as Holling [8], the Resilience Alliance [9] and the Rockefeller Foundation [10] proposed a theoretical framework, which is now relatively mature. In China, the current research is evolving from conceptual summaries, content classification and comparative studies of resilience to practical studies of national conditions. In the 1970s, Holling [8] redefined resilience and applied it to the ecological environment. In this definition, resilience is a system property, and persistence or the probability of extinction is the result [11]. In 1990, resilience was first applied to cities, and its main purpose was to reduce the impact of disturbances on city systems [12,13]. In the 21st century, Holling [14] redefined and extended resilience within the context of CASs as the ability to withstand disturbance, self-organize, and learn and adapt. After that, the concept of resilience with cities as the research core has developed greatly [15]. At present, resilience is understood in three forms: engineering resilience, ecosystem resilience and evolutionary resilience. Engineering resilience emphasizes that a system has one and only one equilibrium state and uses the speed of return to the equilibrium state to characterize the resilience. Most scholars have focused on building models to address extreme weather, disaster recovery and energy consumption [16,17]. Ecosystem resilience focuses on the buffer capacity of a system to resist shocks and maintain functions, and it emphasizes multiple equilibrium states. It is mainly used in species continuation [18,19,20,21], environmental restoration and other fields [22,23,24]. The third type of resilience focuses on the interaction between human society and the ecological environment and emphasizes the importance of the learning ability and self-organization ability of a system in the process of adapting to shocks [25]. The purpose is to improve the ability for continuous evolution of a system; thus, it is called evolutionary resilience. Relevant studies include regional disaster resistance assessment and post-disaster recovery [26,27], resilient city assessment and planning assumptions [6,10,28,29]^,^ etc. It is worth noting that the most typical and widely recognized evolutionary resilience is the adaptive cycle model proposed by Gunderson and Holling [14]. On the whole, most of the research on urban resilience starts from the evaluation of the process of post-disaster reconstruction, from microscopic aspects of the community resilience, urban planning and management, or from building models that are more qualitative than quantitative.

At present, the urban system resilience theory based on adaptive cycles and an emphasis on comprehensive system feedback has dominated urban resilience research. This paper also believes that resilience analysis from the perspective of evolution promotes a clearer understanding of how cities develop stably in a complex environment. Urban resilience defines the relationship between a disturbance and the subject experiencing disturbance from a dynamic perspective. On the one hand, it recognizes the usualness of disturbances and the dynamic balance of the system, while on the other hand, it is recognized that the results of disturbances on systems with different resilience levels may be quite different. A system with a high resilience has a strong ability to maintain its integrity. Therefore, it is necessary to quantitatively evaluate the resilience of urban systems. Urban development is a process of change in social mobility, and its complexity does not simply depend on whether many subsystems and layers occur. The key lies in the complicated relationship between humans, society and cities and the complicated coupling mechanism between cities and the natural ecological environment. Therefore, the assessment of the resilience of the entire urban system should include an assessment of the various components and their relationships and interactions, but there are few existing studies that involve the assessment of the resilience of the entire urban system and consider the relationships and interactions between the various components [12]. In view of the above, based on the CAS theory and adaptive cycle model, this paper builds an urban resilience research framework and evaluation index system from the four aspects of urban systems, including stability, self-organization, learning adaptability and transformability, and comprehensively applies the Technique of Order Preference Similarity to the Ideal Solution (TOPSIS) method, system entropy flow method, spatial analysis method and other analysis methods to examine the spatio-temporal evolution of the urban resilience characteristics of the Lanzhou-Xining urban agglomeration to enrich the relevant theoretical and empirical research on urban resilience.

## 2. Basic Theory

### 2.1. Theory of a Complex Adaptive System

The CAS theory focuses on the analysis of the self-organization mechanism of a system, and holds that the system subject shows dynamic characteristics as well as adaptive and learning processes [30]. An urban system is a CAS in which people and nature are closely connected. It constantly circulates and oscillates in the evolutionary process of prosperity or decline. Such cycles, oscillations and new forms are the result of the interactions between the material, energy and information flows in a city’s internal and external environments at the multilayered space-time scale of a city and are also a manifestation of a city’s continuous evolution process through self-organization and active learning in adaptation to the environment.

### 2.2. Evolutionary Urban Resilience and the Adaptive Cyclic Model

The research on urban resilience is an important dimension to further understand the complex evolutionary process of urban systems. Regardless of whether a CAS is under external disturbance, it exhibits the characteristics of change. Its resilience does not rely on returning to the normal state, but the ability of complex social-ecological systems to change, adapt and transform in the face of pressure and stress has obvious evolutionary characteristics [31]. Evolutionary urban resilience can be regarded as the collection of various abilities of urban systems to achieve adaptive development under pressure disturbances. Generally, urban resilience can be summarized as three aspects of the capacity of urban complex systems [32,33,34]: (1) the ability of a city to maintain normal functions without external help in the event of an emergency (such as earthquake, flood and public health emergencies), (2) the ability of a city to return to normal when a subsystem is disturbed, and (3) the ability of a city to meet future challenges using the experience and basis of structural adjustment and learning adaptation. It is embodied in the four characteristics of the urban system, namely: (1) stability, which is the bearing mechanism of the urban structure, refers to the ability of the system to resist and absorb a disturbance and prevent a change in the basic urban structure; (2) self-organization, which is the dynamic supplementary mechanism of a city as a CAS and refers to the ability of the system to provide the power needed to resolve any impacts; (3) learning adaptability, which is the innovation mechanism of an urban system and refers to the ability of the system to learn and adapt to the environment when dealing with shocks; and (4) transformability, which is the dynamic transformation mechanism of an urban system and refers to the system ability to restore normal operations and reach a new equilibrium state after encountering impacts. These four characteristics, which are independent and can connect smoothly, scientifically and comprehensively explain the entire development process of a resilient city. The specific evaluation of urban resilience can be obtained by synthesizing these four abilities in the process of urban development.

The adaptive cycle model constructed provides an analytical model to explain the evolutionary resilience theory of the CAS. The model proposes an adaptive cycle of the system function and structure evolution with four phases: exploitation(r), conservation(K), release(Ω), and reorganization(α) [14]. The exploitation phase is related to system structure and functional stability. The conservation phase is related to the possible fossilization and degradation of the system in the later phase [35]. With the continuation of the conservation phase, increasingly more resources are locked, the system structure is solidified, the development path becomes fixed, the resilience is reduced, the response of the system to external shocks becomes less flexible, and accidents may occur [36]. This is followed by the release phase of chaotic collapse, in which the system breaks down and gradually gives way to the reorganization phase, during which the system may encounter new opportunities and reallocate resources, with great uncertainty. It is a time of innovation and change, and a time when crisis turns into opportunity. The reorganization phase is followed by a new exploitation phase, whereby the system may experience diversified spiral development [37]. The four stages have different resilience characteristics. Which stage a system is in and which stage it will evolve into depends on the key indicators that affect the basic functions of the system [38,39,40,41] (Figure 1). (1) Exploitation phase: the urban resilience increases rapidly, and the four system characteristics are low, but the increase is significant, especially the stability and self-organization. In this phase, a city continues to accumulate resources and improve its competitiveness. (2) Conservation phase: the urban resilience and system attributes are relatively high, increase slowly, and even exhibit a downward trend. In this phase, most of the resources are stored and utilized, the path certainty and system resilience of urban development are high and close to the threshold, the system development level tends to mature, and the risk of structural solidification and path locking appears. (3) Release phase: the urban resilience and various attributes show an obvious downward trend, especially a rapid decline in self-organization. The system enters a chaotic collapse period, and urban problems become prominent. (4) Reorganization phase: the resilience and attributes are on the rise, especially the self-organization and learning adaptability. The system is in a period of innovation and structural adjustment, with great uncertainty.

### 2.3. Research Framework

Based on the perspective of resilience research of CASs and Holling’s adaptive cycle model, this paper proposes an open research framework of urban resilience with multi-dimensional and multi-factor coupling (Figure 2), in an attempt to provide a powerful tool for the systematic study of urban resilience. Based on the abstract analysis of complex adaptive characteristics and adaptive cycle characteristics of urban systems, this research framework concretized four attributes of urban resilience and then constructed an open index system to measure urban resilience, taking the Lanzhou-Xining urban agglomeration as an example to conduct an empirical study. The Lanzhou-Xining urban agglomeration is an important inter-provincial urban agglomeration in western China. As an important part of China’s Silk Road Economic Belt and one of the few leading economic regions in western China, what are the spatial-temporal characteristics of its resilience? What are the characteristics of each city’s evolutionary resilience? What types of urban development have been formed based on these characteristics, and how can adaptive governance be conducted in each city? These are the scientific issues that this paper intends to discuss.

## 3. Research Design

### 3.1. Index System

To improve the resilience of CASs, some scholars have proposed seven principles of sustainable management and governance [42,43]. Based on these principles, this study combined with the characteristics of the study area, selects indicators from four aspects of system stability, self-organization, learning adaptability and transformability, and analyzes the spatial-temporal evolution of urban resilience in the Lanzhou-Xining urban agglomeration (Table 1).

Stability: This is the ability of economic, social, environmental and other subsystems in the urban system to interact, permeate and interweave with each other to ensure that the main function of the system is not affected. Referring to the “social-economy-ecology” resilience evaluation system constructed by previous scholars [44,45], this paper also selects stability indicators from three areas: the social system, economic system and environmental system. The social system mainly provides convenient living conditions and comfortable living environment for citizens, including medical, transportation, education, public services and other basic aspects. The economic system is mainly reflected in whether the economic base can support the operation of the urban superstructure. The environmental system ensures that city activities can be carried out in a good ecological environment, which includes the vegetation growth status and the degree of interference of human activities on the environment.

Self-organization: This is the dynamic mechanism of urban development. With the accelerated development of globalization and information technology, cities are more closely connected, and the self-organization of cities is more reflected in the flow space of multiple factors such as population, goods, capital, technology and information, which has an impact on the spatial development mode of cities or regions [46,47]. Therefore, evaluation indices are selected from the material, energy and information of an urban system, including the change in population, the distribution of resources, the amount of investment and the mastery of information.

Learning adaptability: This reflects the ability of actors in a system to influence resilience. Establishing a learning society enables the city system to adapt in line with the law of development of science and technology of the national innovation system with Chinese characteristics [48]. Therefore, this study uses learning innovation ability as a measure of the city’s ability to adapt to change. We establish three factor levels of knowledge: innovation, technological innovation and basic innovation environment. These include indicators related to universities, enterprises, government and residents.

Transformability: This reflects the ability of cities to optimize their structure and create “new systems” when making adaptation adjustments. The two key characteristics of transformation ability are (1) to destroy and dismantle the existing system actively and (2) to create and establish feasible alternatives [49]. Moderate urban scale and excellent urban structure are effective means to solve acute shock and chronic pressure in cities [50]. Therefore, we consider selecting indicators to measure the transformability of cities by selecting urban connectivity, urban form and facility density.

### 3.2. Mathematical Analysis Model

#### 3.2.1. Calculation of the Urban Stability

For the stability calculation, we apply the TOPSIS method, which is a common decision method in multiobjective decision analysis of finite schemes in system engineering and systematically analyzes the gap between the system stability and its ideal state [51].

① The evaluation matrix *Y* is listed:(1)$$Y=\left[\begin{array}{cccc} y_{11} & y_{12} & \cdots & y_{1n}\\ y_{21} & y_{22} & \cdots & y_{2n}\\ \vdots & \vdots & \vdots & \vdots \\ y_{m1} & y_{m2} & \cdots & y_{mn} \end{array}\right]=\left[\begin{array}{cccc} r_{11}\cdot \omega_{1} & r_{12}\cdot \omega_{2} & \cdots & r_{1n}\cdot \omega_{n}\\ r_{21}\cdot \omega_{1} & r_{22}\cdot \omega_{2} & \cdots & r_{2n}\cdot \omega_{n}\\ \vdots & \vdots & \vdots & \vdots \\ r_{m1}\cdot \omega_{1} & r_{m2}\cdot \omega_{1} & \cdots & r_{mn}\cdot \omega_{n} \end{array}\right]$$
where *r_mn_* is the nth index in the *m*th year after standardization, and *ω* is the weight.

② The positive and negative ideal solutions are determined:(2)$$Y^{+}=\max\left\{\left(y_{ij}|i=1,2,\cdots ,m\right)\right\}\left(j=1,2,\cdots n\right)=\left\{y_{1}^{+},y_{2}^{+},\cdots ,y_{n}^{+}\right\}$$
(3)$$Y^{-}=\min\left\{\left(y_{ij}|i=1,2,\cdots ,m\right)\right\}\left(j=1,2,\cdots n\right)=\left\{y^{-},y_{2}^{-},\cdots ,y_{n}^{-}\right\}$$
where *Y^+^* is the maximum value of the *j*th index in year *i* and *Y^+^* is considered as the positive ideal solution, based on which the most ideal scheme can be selected; *Y^−^* is the minimum value of the *j*th index in year *i*, and *Y^−^* is the negative ideal solution, i.e., the worst scheme.

③ The distance between each index and the positive and negative ideal solutions is calculated:(4)$$D_{j}^{+}=\sqrt{\sum_{i=1}^{m}{\left(y_{j}^{+}-y_{ij}\right)}^{2}}$$
(5)$$D_{j}^{-}=\sqrt{\sum_{i=1}^{m}{\left(y_{j}^{-}-y_{ij}\right)}^{2}}$$
where *D_j_^+^* is the distance between the *j*th index and *y_j_^+^*, and *D_j_^−^* is the distance between the *j*th index and *y_j_^−^*.

④ The closeness degree is calculated:(6)$$C_{j}=\frac{D_{j}^{-}}{D_{j}^{+}+D_{j}^{-}}$$
where *C_j_* is between 0 and 1, which comprehensively reflects the stable state of the evaluation object reflected by the distance indexes *D_j_^+^* and *D_j_^−^*. The larger the *C_j_* value is, the closer the evaluation object is to the ideal solution, and the better the stability of the urban system.

#### 3.2.2. Calculation of the Urban Self-Organization

This study adopts the system entropy flow method to measure the dissipative structure characteristics of the system to represent the self-organization of a city [52]. The higher the entropy flow value is, the more energy, matter and information are exchanged with the external environment, the stronger its aggregation and radiation capacity, and the stronger its urban resilience. The calculation formula is as follows:(7)$$S=\sum_{j=1}^{m}\left[{\left(-1\right)}^{n}\frac{\left|r_{j}-r_{1}\right|}{r_{1}}\omega_{j}\right]$$
where *S* is the sum of the entropy flows generated by the system, *ω_j_* is the weight of index *j*, which is calculated by the entropy weight method, and (−1)*^n^* is a symbolic function. When the index exceeds the initial state, *n* is 1, indicating the input of a negative entropy flow, and otherwise, *n* is 2. Moreover, *r_j_* is the *j*th index, and *r*_1_ is the initial value.

#### 3.2.3. Calculation of the Urban Learning Adaptability

Due to the different intellectual capital and innovation ability levels of each city, differences in governance, technology and planning occur. Different cities have different abilities to adapt to new or unexpected disturbances. The greater the urban learning adaptability is, the stronger its resilience is [53]. The calculation formula is as follows:(8)$$L=\sum_{j=1}^{n}r_{j}\cdot \omega_{j}$$
where *L* is the learning adaptability index, *r_j_* is the *j*th index, and *ω_j_* is the weight of index *j*.

#### 3.2.4. Calculation of the Urban Transformability

An urban system can implement structural adjustments through its transformability and realize the transformation of its socioeconomic status. The openness and dynamic characteristics of urban systems determine that the transformability is the fundamental driving force for their development and evolution. The calculation formula is as follows:(9)$$T=\sum_{j=1}^{n}r_{j}\cdot \omega_{j}$$
where *T* is the transformability index, *r_j_* is the *j*th index, and *ω_j_* is the weight of index *j*.

#### 3.2.5. Calculation of the Urban Resilience

An urban system is a complex organic whole, and its resilience is the comprehensive result of stability, self-organization, learning adaptability and transformability. The calculation formula is as follows:(10)$$R=\omega_{1}C+\omega_{2}\left|S\right|+\omega_{3}L+\omega_{4}T$$
where *R* is the resilience, *C* is the stability, *S* is self-organization, *L* is the learning adaptability, *T* is the transformability, and *ω*_1_, *ω*_2_, *ω*_3_, and *ω*_4_ refer to the weight of four attributes, respectively.

### 3.3. Overview of the Study Area

The Lanzhou-Xining urban agglomeration, located in Northwest China, is an important support for the China-Central Asia-West Asia Economic Corridor and an important inter-provincial urban agglomeration in Western China. Its populations and towns are relatively dense, and the ecological environment is sensitive and fragile. Since ancient times, it has been a strategic place to safeguard China’s land and ecological security. The planning scope of the Lanzhou-Xining urban agglomeration includes Lanzhou, Baiyin, Dingxi, Linxia, Xining, Haidong, Haibei, Hainan and Huangnan, a total of nine cities with a total area of 179,700 square kilometers. As of the end of 2017, the total population of the whole region was 15.43 million, and the GDP per capita was 35,902 yuan. According to The Ranking of Chinese Urban Development Potential, compared with the other 18 urban agglomerations in China, the development speed of the Lanzhou-Xining urban agglomeration is lower. The development potential index of the Lanzhou-Xining urban agglomerations is 52, which is in the embryonic stage of development. It is regional urban agglomerations that the state has focused on cultivating in recent years. As an important part of China’s Silk Road Economic Belt and one of the few leading economic regions in western China, the Lanzhou-Xining urban agglomeration has a special topography and an important geographical location, which is of profound significance for strengthening ethnic unity and stabilizing the northwest and southwest regions (Figure 3).

### 3.4. The Data Source

The social and economic data were obtained from statistical yearbooks, statistical bulletins of national economic and social development and enterprise statistics. Remote sensing image data were obtained from data Center for Resources and Environmental Sciences, Chinese Academy of Sciences (https://www.resdc.cn) (accessed on 20 April 2020).

## 4. Results

### 4.1. Analysis of the Urban Stability, Self-Organization, Learning Adaptability and Transformability

Selecting nine cities in the Lanzhou-Xining urban agglomeration as research objects, the stability, self-organization, learning adaptability and transformability of each city are calculated by the methods described above (Figure 4).

(1)Stability: From 2010 to 2017, there were large fluctuations in stability and differences between the cities. Among them, the stability of Xining showed first a rising and then declining trend, with a maximum decline of 50%. The stability of Baiyin increased from 0.23 in 2010 to 0.66 in 2017, a nearly two-fold increase, whereas the stability of Lanzhou changed by only 2%, with the most gradual trend. The fundamental reason is that the cities have begun to transform and develop by means of improving development quality and controlling development speed, and the original stable situation has been disrupted. For example, the GDP growth rate of Xining has reduced from 18.2% to 10.8% in the past eight years. Baiyin has vigorously developed tertiary industry (32% to 47%), and increased the proportion of environmental protection expenditure (12.42%), striving to achieve sustainable development. In recent years, the industrial structure of Lanzhou has been solidified and the development path has become fixed, which has led to a slow rise in the urban economic and social development levels, high stability and lack of obvious change. Generally speaking, the city has high stability and a strong ability to resist disturbance, and the urban system does not easily collapse. However, the high stability may lead to structure solidification, development path locking, and the city entering a long period of slow development or even stagnation, which is not conducive to the adaptive cycle development of the city.(2)Self-organization: From 2010 to 2017, the self-organization of each city exhibited an upward trend. That of Haidong increased the fastest and that of Lanzhou increased the slowest. In 2017, the entropy value of Haidong reached 2.15, which was mainly a benefit of the opening of the Lanzhou–Xinjiang high-speed railway, which enables a large number of people, resources, funds and information to flow in and out of the city. In 2017, Lanzhou’s self-organization index was 0.5 and has increased slowly over the years. The main reason is that although the total amount of material and energy in the city is large, the growth rate is not high. Over the past eight years, the population density has increased by 3.06%, the investment in fixed assets in the whole society has increased by 99.09%, and the number of Internet users has increased by 187.93%, which is much smaller than the same index in Haidong (23.12%, 628.37% and 600.45%, respectively). Generally, the higher the self-organization of a city is, the higher its orderliness and the more conducive this is to its sustainable development. However, if the vitality of urban development is insufficient and the structure has solidified, urban development will be slow. At this time, it is necessary to intervene in the internal and external urban environments using various stimuli and development opportunities to generate new vitality, break through the existing lock, and enter a higher level, more complex and more orderly development stage.(3)Learning adaptability: From 2010 to 2017, the learning adaptability of each city exhibited the obvious Matthew effect and regional polarization phenomenon in space. Lanzhou had the highest learning adaptability (a mean value of 0.94), followed by Xining (a mean value of 0.35), and the remaining seven cities exhibited a low learning adaptability. The reason for this phenomenon is that the provincial capital city has concentrated many of the higher education resources of the whole province (Lanzhou has 32 ordinary colleges and universities, while Xining has 10 ordinary colleges and universities), and in addition to the gathering of higher education resources, high-tech industries and high-tech personnel with a high innovation ability have converged onto the provincial capital city (the number of R&D personnel in Lanzhou is 22,742, and the number of R&D personnel in Xining is 2701). The knowledge and talent in the capital city are constantly accumulating, and together with a good base for urban economy (the per capita disposable income of urban residents in Lanzhou was CNY 32,331; the per capita disposable income of urban residents in Xining was CNY 30,043), result in a high urban learning adaptability. The high urban learning adaptability is not only conducive to the upgrading and transformation of industries and adjustment of the urban industrial structure but is also conducive to the rapid transition of the urban system from the conservation phase into the release phase and thereafter into the reorganization phase, thereby leading to a higher-level exploitation phase.(4)Transformability: From 2010 to 2017, the transformability of each city reflected an obvious hierarchy (high, moderate and low levels). Lanzhou and Xining maintained high values for many years, with average values of 0.81 and 0.89, respectively. Baiyin, Haidong, Dingxi and Linxia were at moderate levels, and Haibei, Hainan and Huangnan were at low levels. Lanzhou and Xining had the lowest traffic accessibility index (the average value was 7 and 6.67, respectively), but their road network density was relatively high (the average value was 0.6 and 0.96 km/km^2^, respectively), and their density of industrial enterprises above the designated size was the highest (the average value was 271/km^2^ and 351/km^2^, respectively). When unknown interference occurs, cities have a strong ability to introduce or allow new variables to create a new stable environment. The higher the urban transformability is, the greater the external contact and the higher the degree of openness, which better promotes the external economic development of the city. Moreover, the opportunity for external development can be utilized, thus promoting the restructuring of the urban structure, producing a cross-scale effect on the urban development cycle, and realizing the city’s transformation and development.

### 4.2. Urban Resilience and Adaptive Cycle Analysis

According to the results of each urban resilience attribute, this paper calculates the resilience indexes of each city in the Lanzhou-Xining urban agglomeration (Figure 5 and Figure 6). Combined with the evolutionary urban resilience model, this study divides the nine cities into four types: exploitation–reorganization, conservation–release, conservation–exploitation and exploitation.

Exploitation–Reorganization: Haidong. The city directly enters from the exploitation phase with a low resilience into the reorganization phase with a high resilience, thus realizing leapfrog development. From 2010 to 2017, Haidong experienced the fastest growth in resilience, from 0.82 in 2010 to 3.23 in 2017. The resilience in 2014 (1.92) was higher than that of the provincial capital Xining (1.87), and its resilience in 2016 (2.82) was higher than that of the provincial capital Lanzhou (2.61). The growth rate over the past eight years reached 294.15%, and thereby it became the city with the highest resilience in the whole urban agglomeration. This leapfrog growth benefited from the construction of a high-speed railway (the Lanzhou–Xinjiang high-speed railway) with a high influx of population, resources, capital and information into Haidong, which not only changed the economic development structure of the city but also quickly filled urban infrastructure vacancies, resulting in the city attaining a sufficient resistance and coordination adaptability in the face of disturbance.

Conservation–Release: Lanzhou. The city’s urban resilience approaches the threshold, its growth is slow, path locking is obvious, and the city is in the high-resilience stage. In addition, the system appears to experience the momentum of collapse and release. From 2010 to 2017, Lanzhou attained a high stability and self-organization level, thereby showing a gradual upward trend. Its learning adaptability was the highest, with little change, and its transformability was high, whereas the decline was obvious. The urban resilience increased slowly, increasing by 12.84% over the past eight years. The urban system developed slowly as most resources were stored and used. The urban system structure remained stable with a high level of certainty, and the urban development vitality was not high. Therefore, it is urgent to adjust the industrial structure of Lanzhou to promote the urban system from the conservation phase to the release phase and into a higher level of development to realize the green, harmonious and transformative development of the city.

Conservation–Exploitation: Xining. The city’s urban resilience is high and its growth is obvious (47.75%). Its stability is greatly reduced, the transformability of the city has notably increased, and the urban system exhibits a development trend from the conservation phase with a high resilience to the exploitation phase at a higher level. However, the self-organization of Xining slowly increases, and its learning adaptability is low, which hinders the transformation and development of the city.

Exploitation: Baiyin, Dingxi, Linxia, Haibei, Hainan, and Huangnan. The urban resilience is not high, but it is increasing rapidly. The cities are in the stage of rapid development, and the available resources are rapidly accumulating. The system diversity and connectivity are constantly improving, and the competitiveness is rapidly enhancing. Over the past eight years, the resilience of each city has rapidly increased (the resilience of Baiyin has increased by 174.68%, that of Dingxi by 181.66%, that of Linxia by 182.33%, that of Haibei by 125.28%, that of Hainan by 239.6% and that of Huangnan by 153.37%). The urbanization rate of these cities is low (in 2017, the urbanization rate of Baiyin was 49.32%, that of Dingxi was 34.33%, that of Linxia was 34.47%, that of Haibei was 30.12%, that of Hainan was 30.13%, and that of Huangnan was 28.52%). These cities have low learning adaptability and transformability, stable structure, and are basically at a low level of balanced development stage, but they have certain development potential.

### 4.3. Countermeasures for Adaptive Governance for the Various Cities

Exploitation–Reorganization: Haidong. The stability and self-organization of Haidong increase notably, but its learning adaptability and transformability are relatively low. Therefore, Haidong should give full play to the location advantages of the key cities in the Silk Road Economic Belt and the gateway cities on the Qinghai Tibet Plateau, strengthen external relations, and attract more people, resources and capital inflow. Simultaneously, the city should pay close attention to infrastructure construction and improve the capacity of the basic environment for innovation, increase the investment in science and technology, improve the quality of education, train more scientific and technological talents, improve the learning adaptability and transformability of the urban system, and promote the high-quality development of the city.

Conservation–Release: Lanzhou. The urban resilience of Lanzhou approaches the threshold value, path locking is obvious, and the vitality of urban development is not high. Therefore, Lanzhou should fully utilize the agglomeration characteristics of its higher education resources and high-tech enterprises, give full play to the existing economic advantages, accelerate the adjustment of its industrial structure, vigorously develop the traditional Chinese medicine, tourism and cultural industries, promote the transformation and upgrading of traditional industries, strengthen the construction of ecological communities, increase investments in infrastructure such as subway construction and new area construction, and raise the upper limit of urban development to promote the urban system from the conservation phase to the release phase and into the reorganization phase. The city can thereby realize its green, harmonious development, enhance the surrounding cities and drive the development of the whole urban agglomeration.

Conservation–Exploitation: Xining. The city’s self-organization slowly increases, and its learning adaptability is low, which is not conducive to the transformation and development of the city to a higher stage. Therefore, Xining should increase the investments in science and technology, attract more high-level talents, increase the investments in education, cultivate more scientific and technological talents, vigorously promote the construction of transportation, communication infrastructure and ecological communities, greatly support the development of the tourism and cultural industries, and prevent the obvious path locking phenomenon in urban development.

Exploitation (growth): Baiyin, Dingxi, Linxia, Haibei, Hainan, and Huangnan. The first three cities belong to Gansu Province with flat terrain, abundant tourism resources, and a large proportion of the tertiary industry. Therefore, first, on the basis of continuing to develop the tertiary industry, these cities should strengthen the construction of the infrastructure such as transportation and communication, utilize the favorable opportunities of western and high-quality development of the Yellow River according to the western and Yellow River Basin policies, respectively, and develop suitable industries, such as ecological agriculture, tourism, and traditional Chinese medicine. Second, these cities should increase the financial expenditure on education, cultivate high-tech talents, and improve the urban learning adaptability and transformation ability. Finally, these cities need to take advantage of the trickle-down effect of provincial capitals and try to absorb external population, resources and funds to enhance their vitality and resilience. The latter three cities belong to Qinghai Province, with a high altitude, a large proportion of ethnic minorities, and with agriculture and animal husbandry as the main economic means of production. Therefore, they should first promote urbanization by increasing urban jobs, expanding infrastructure construction, and increasing social welfare. Second, they should fully utilize the mineral and tourism resources to improve the income levels, build urban resilience, and enhance their sustainable development capability.

## 5. Discussion

The evolutionary resilience model used in this paper is based on the spiral evolution process of the whole urban system, which provides an effective and comprehensive resilience evaluation tool for the city to cope with disturbances. Although this study was conducted on the Lanzhou-Xining urban agglomeration from 2010 to 2017, the diversity of urban forms can provide a reference for the study of urban resilience in other regions, so the results of this study can be verified by similar studies in other regions. For example, Lu et al. [54] conducted an empirical study on the resilience of 31 provincial capital cities in China and found that they have higher adaptability in the face of external shocks. Yang [55] compared the economic development status of 31 provinces in China and pointed out that in the great opportunity of the western development strategy, it is completely possible for Qinghai Province to leapfrog some traditional development stages and realize local development by leaps and bounds. Other recent studies on the regional development of the Lanzhou-Xining urban agglomeration have confirmed the results of this study. For example, Zhang et al. [56] found through an analysis of the economic–social–ecological coupling coordination of the Lanzhou-Xining urban agglomeration in recent years that the land use efficiency of the core area of the urban agglomeration is significantly higher than that of the surrounding areas. Zhong et al. [57] analyzed the spatio-temporal evolution of the industrial structure level of the Lanzhou-Xining urban agglomeration and found that by 2019, the dual core region composed of Lanzhou and Xining has always maintained a high level of industrial technology and innovation capacity.

This study has focused on the dynamic resilience of urban social-ecological systems and less on the static resilience of engineering structures. The method of resilience evaluation could be improved and made more accurate by enriching the indicators of the evaluation system and improving the comprehensive calculation model of urban resilience. In addition, the Lanzhou-Xining urban agglomeration is located in the core area of the Silk Road Economic Belt, which is of great importance in the study of urban sustainable development. Moreover, due to its topography, urban expansion is limited, which is both typical and unique. Therefore, other urban agglomerations could be analyzed in the future to obtain more universal conclusions and insights.

## 6. Conclusions

From the perspective of CAS theory, this paper analyzes through a literature review the basic dynamic mechanisms important to the evolution of urban systems. Combined with Holling’s definition of resilience and adaptive cycle model, urban resilience is understood as the comprehensive results of four basic attributes in an urban system: stability, self-organization, learning adaptability and transformability. A multi-level open index evaluation system is then constructed. An evolutionary urban resilience analysis framework is proposed to quantitatively analyze the spatial and temporal characteristics of the urban resilience of the Lanzhou-Xining urban agglomeration from 2010 to 2017. The research conclusions are as follows:(1)The research perspective of the evolutionary urban resilience and open index system provide an analytical framework based on the CAS theory for the analysis of the urban resilience of the Lanzhou-Xining urban agglomeration. In this framework, the stability, self-organization, learning adaptability and transformability of urban systems jointly determine the magnitude of urban resilience and its adaptive evolution stage. The adaptive cycle of urban system evolution is embodied in the Lanzhou urban agglomeration.(2)From 2010 to 2017, the urban system attributes of the Lanzhou-Xining urban agglomeration present obvious differences. The stability of each city has large fluctuations and differences. The self-organization and the order of the urban system are increasing. The trend in learning adaptability is stable, exhibiting the obvious phenomena of the Matthew effect and regional polarization in space. The transformability reveals a distinct hierarchy (high, moderate and low levels).(3)The urban resilience of the Lanzhou-Xining urban agglomeration is increasing as a whole, with the spatial evolution from double cores (Lanzhou and Xining) to three cores (Lanzhou, Xining and Haidong) and the polycentric mode. This process corresponds to the adaptive cycle development process of the urban system and the Chinese new urbanization development strategy over time and space. Through adaptive cycle development, the urban resilience of the Lanzhou-Xining urban agglomeration is constantly improved, and the capability for urban sustainable development is gradually enhanced.(4)According to the spatio-temporal characteristics of urban resilience and its attributes, combined with the adaptive cycle model of the urban system from the perspective of evolutionary resilience, the cities of the Lanzhou-Xining urban agglomeration are divided into four types, namely, exploitation–reorganization (Haidong), conservation–release (Lanzhou), conservation–exploitation (Xining) and exploitation (the other 6 cities) cities, and according to the characteristics of the various cities, corresponding adaptive countermeasures are proposed. This research on the urban resilience of the Lanzhou-Xining urban agglomeration verifies the applicability of the research and analysis framework at the meso-level, and proposes further research on urban resilience based on the theory of CAS to provide a theoretical reference for the research on the spatio-temporal evolution of urban resilience.

## Figures and Tables

**Figure 1 ijerph-19-13667-f001:**
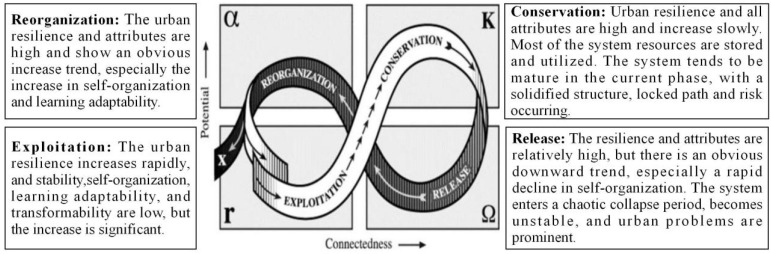
Evolutionary urban resilience model (Refer to the adaptive cycle model of Holling and Gunderson [14]).

**Figure 2 ijerph-19-13667-f002:**
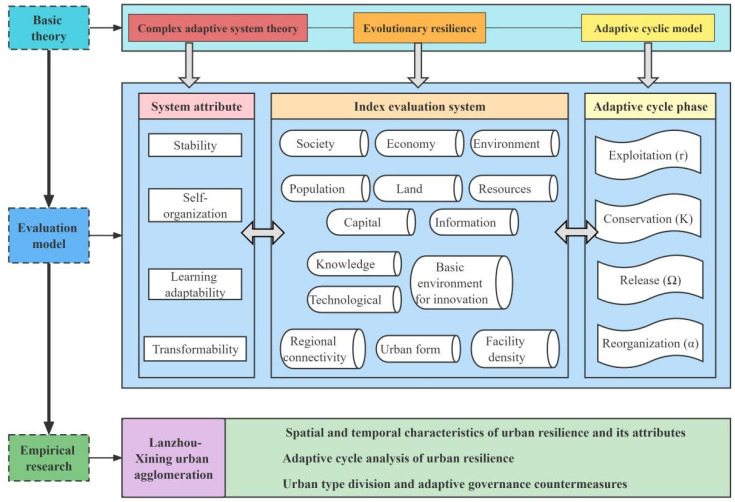
The analytical framework of evolutionary urban resilience.

**Figure 3 ijerph-19-13667-f003:**
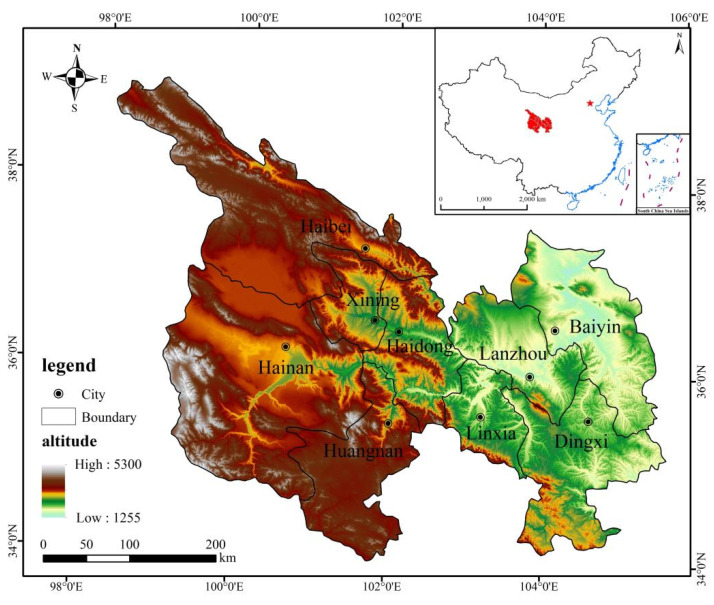
Study area. Note: This map is based on the standard map of the National Administration of Surveying, Mapping and Geographic Information, and the base map is not modified.

**Figure 4 ijerph-19-13667-f004:**
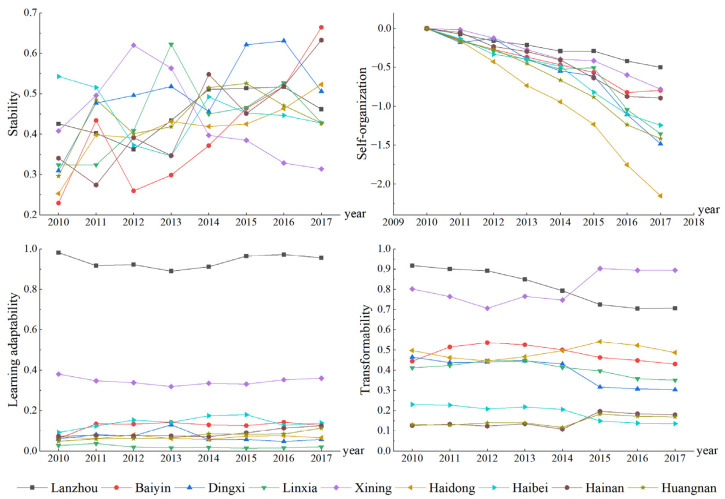
Four characteristic index trends of nine cities from 2010 to 2017.

**Figure 5 ijerph-19-13667-f005:**
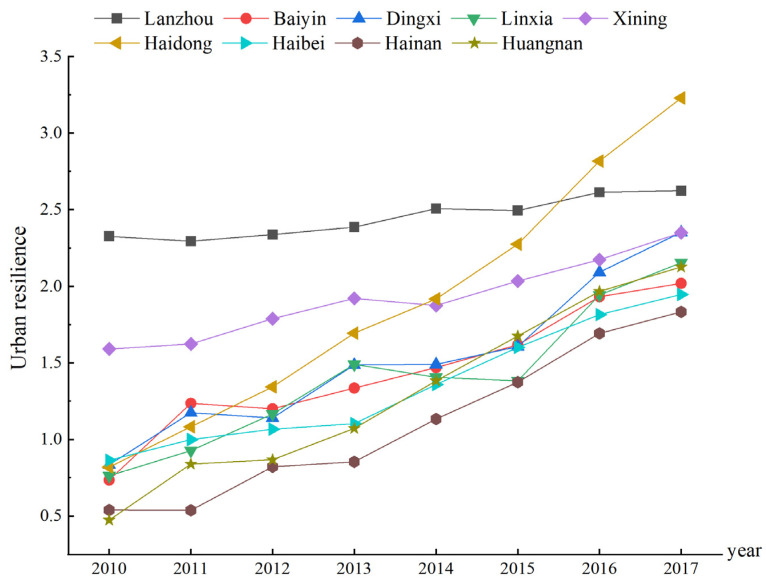
Trends in urban resilience of the nine cities from 2010 to 2017.

**Figure 6 ijerph-19-13667-f006:**
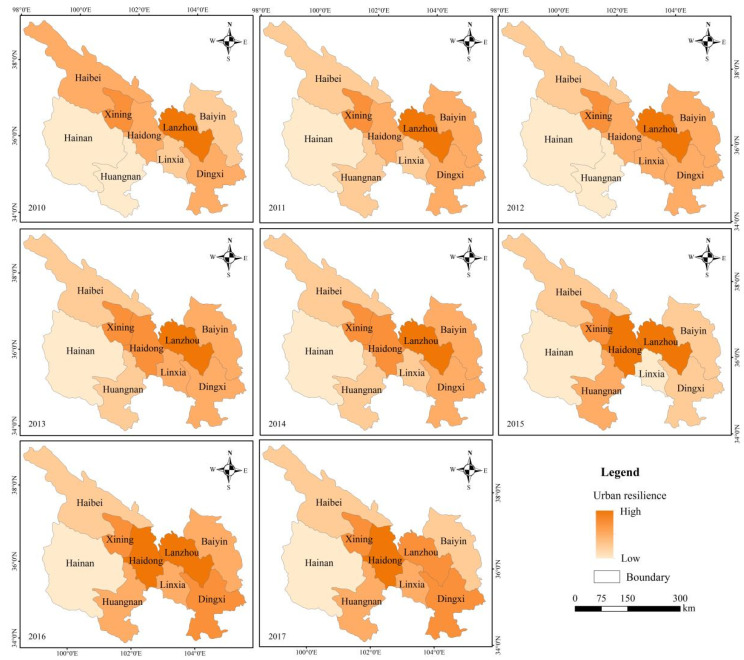
Spatial distribution of the urban resilience of the nine cities from 2010 to 2017. Note: This map is based on the standard map of the National Administration of Surveying, Mapping and Geographic Information, and the base map is not modified.

**Table 1 ijerph-19-13667-t001:** The index evaluation system of the urban resilience.

Target Layer	Criterion Layer	Factor Layer	Index Layer
Urban Resilience	Stability	Society	Beds per 10,000 people, Public vehicles per 10,000 people, Urbanization rate, Proportion of the education expenditure in the fiscal expenditure, Proportion of the public services expenditure in the fiscal expenditure, and Proportions of the urban and rural populations receiving subsistence allowances in the total population.
		Economy	Proportion of the added value of the primary, secondary and tertiary industries in the GDP and GDP growth rate.
		Environment	Regional vegetation coverage, Proportion of the environmental protection expenditure in the fiscal expenditure, Good-weather frequency, Urban domestic sewage treatment rate, Discharge standard rate of industrial wastewater, and Comprehensive utilization rate of industrial solid waste.
	Self-organization	Population	Population natural growth rate, Population density, and Registered urban unemployment rate.
		Land	Per capita cultivated land area and Per capita construction land area.
		Resources	Per capita mineral resources and Per capita daily domestic water consumption.
		Capital	Investment in fixed assets of the whole society.
		Information	Number of mobile phone users and Number of Internet users.
	Learning adaptability	Knowledge	College students per 10,000 people and Full-time teachers in colleges and universities per 10,000 people.
		Technological	Number of R&D personnel in enterprises and Internal expenditure in enterprise R&D funds.
		Basic environment for innovation	Per capita books in public libraries, Per capita disposable income of urban residents, Per capita disposable income of rural residents, and Proportion of science and technology expenditure in the fiscal expenditure.
	Transformability	Regional connectivity	Regional road network density and Traffic accessibility index.
		Urban form	Urban elongation.
		Facility density	Pipe density and Density of enterprises above the designated size.

## Data Availability

The data supporting reported results can be found at https://data.cnki.net and https://www.resdc.cn.

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
