# Peer review of "Study on Urban Resilience from the Perspective of the Complex Adaptive System Theory: A Case Study of the Lanzhou-Xining Urban Agglomeration"

_ijerph, 2022, doi:10.3390/ijerph192013667_

Round 1
Reviewer 1 Report
review comment:
As a new idea of urban risk management, urban resilience refers to improving the ability of urban system to organize itself, coordinate functions and adapt to uncertainties. How to operate and maintain resilience in the face of increasing and changing risks and challenges, and how to build resilient cities, has become a new topic that needs to be studied urgently in many disciplines. Based on the complex adaptive system and adaptive cycle model, this paper uses four attributes of the urban system to comprehensively represent the resilience index, classifies and analyzes the development stage of each city in the Lanzhou-Xining urban agglomeration, and puts forward future development countermeasures. The topic selection of this paper is of great significance. and the article present us with a t the following problems need to be further modified:
1. The abstract should briefly state the purpose and significance of the research.
2. The explanation of theoretical basis is too lengthy, and it is not clear what is the difference between evolutionary resilience and evolutionary urban resilience. It is suggested to merge 2.2 and 2.3.
3. The overall description of the analysis framework in 3.1 is unclear. It is suggested that this part be described together with the theoretical basis.
4. Why the paper selects indicators from four aspects: stability, self-organization, learning adaptability and transformability? It is suggested to supplement the selection basis of indicators.
5. The author takes the Lanzhou-Xining urban agglomeration as a case for the study. It is unclear what problems or pressures exist in the urban development of this region.
6. The discussion part is relatively weak. It is suggested to highlight the discussion on the uncertainty of the main results of the paper and the comparison with existing studies.
Reviewer 2 Report
please find the comment.
